# Correlation of Biomarkers of Endothelial Injury and Inflammation to Outcome in Hospitalized COVID-19 Patients

**DOI:** 10.3390/jcm11247436

**Published:** 2022-12-15

**Authors:** Levy Munguía, Nayelli Nájera, Felipe de Jesús Martínez, Dylan Díaz-Chiguer, Fiacro Jiménez-Ponce, Miguel Ortiz-Flores, Francisco Villarreal, Guillermo Ceballos

**Affiliations:** 1Dirección Normativa de Salud, Instituto de Seguridad y Servicios Sociales para los Trabajadores del Estado, Mexico City 06030, Mexico; 2Sección de Estudios de Posgrado e Investigación, Escuela Superior de Medicina, Instituto Politécnico Nacional, Mexico City 11340, Mexico; 3Hospital General de Tláhuac, Instituto de Seguridad y Servicios Sociales para los Trabajadores del Estado, Mexico City 06030, Mexico; 4Hospital General de México “Dr. Eduardo Liceaga”, Mexico City 06720, Mexico; 5School of Medicine, University of California San Diego, La Jolla, CA 92093, USA

**Keywords:** COVID-19, inflammatory biomarkers, endothelial injury, syndecan-1

## Abstract

COVID-19 can trigger an intense systemic inflammation and prothrombotic state, leading to a rapid and disproportionate deterioration of lung function. An effective screening tool is essential to identify the patients at risk for severe disease. This observational study was conducted on hospitalized patients with moderate and severe COVID-19 pneumonia in a general hospital in Mexico City between 1 March 2021 and 15 March 2021. Serum samples were analyzed to explore the role of biomarkers of inflammation, coagulation, oxidative stress, and endothelial damage with the severity of the disease. Our results demonstrated that Syndecan-1 and nitrites/nitrates showed a high correlation in severely ill patients. In conclusion, COVID-19 patients with elevated levels of SDC-1 were associated with severe disease. This molecule can potentially be used as a marker for the progression or severity of COVID-19. Preservation of glycocalyx integrity may be a potential treatment for COVID-19.

## 1. Introduction

Severe acute respiratory syndrome coronavirus 2 (SARS-CoV-2) causes coronavirus disease 2019 (COVID-19); clinical manifestations are not specific but shared with viral illnesses [1]. The most common symptoms after viral incubation between 4 and 14 days are cough, fever, fatigue, anosmia, dysgeusia, headache, and sometimes nausea and diarrhea. Clinical manifestations can range from mild to very severe and even fulminant disease [2].

The leading cause of mortality in patients with COVID-19 is a respiratory failure caused by acute respiratory distress syndrome (ARDS) [3]. Several proofs suggest that diverse dysregulations of endothelial cells (ECs), such as alteration of vascular barrier integrity, promotion of a pro-coagulative state, induction of vascular inflammation (endothelitis), and mediation of inflammatory cell infiltration, contribute to the initiation and propagation of ARDS [4].

Mechanisms involved in this endothelial dysfunction are a consequence of excessive cytokine production and the host’s exacerbated pro-inflammatory response, particularly by the actions of IL-6 and TNFα, which are increased in critically ill patients [5]. Although cytokines exert their pathogenic effects through the mechanisms related to endothelial dysfunction, IL-6 increases vascular permeability and promotes the secretion of pro-inflammatory cytokines by endothelial cells, amplifying cytokine release. TNFα activates glucuronidases that degrade endothelial glycocalyx, releasing glycoproteins such as syndecan, glypican-1, and heparan sulfate, and upregulating hyaluronic acid synthase 2, resulting in a deposition of hyaluronic acid in the extracellular matrix [5].

Furthermore, a severe feature of COVID-19 is the activation of coagulation pathways with the potential to spread intravascular clots related to disruption of vascular integrity and ECs death. Overall, the exposure of the thrombogenic basement membrane results in the activation of the clotting cascade [5]. Moreover, ECs activated by IL-1β, and TNFα initiates coagulation by expressing P-selectin, von Willebrand factor, and fibrinogen, to which platelets bind. In turn, ECs release trophic cytokines that further augment platelet production. The body mounts countermeasures to dissolve fibrin-rich blood clots, producing high fibrin breakdown products (D-dimers) [5].

During endothelial dysfunction, several structures are damaged, particularly the glycocalyx, a mediator for cell–cell interactions which protects the cell membrane from the direct action of physical forces and stresses, allowing the membrane to maintain its integrity. The glycocalyx plays a critical role in the development and progression of many diseases [6].

Syndecan-1 (SDC-1) is the core protein in heparan sulfate proteoglycan expressed in endothelial cells and the primary marker of glycocalyx degradation. An elevated serum level of SDC-1 is associated with endothelial injury in cardiovascular disease, diabetes, and sepsis [7,8]. Endothelial dysfunction leads to poor outcomes in COVID-19 patients by promoting inflammation, coagulation, and leukocyte infiltration [9].

This study aimed to elucidate whether the biomarkers of inflammation and endothelial damage are related to the clinical severity and outcome of patients with COVID-19 and to identify patients who may develop long-term consequences and require close clinical monitoring in rehabilitation.

## 2. Materials and Methods

The present study is a prospective observational study carried out at Tlahuac General Hospital of the Institute for Social Security and Services for State Workers in Mexico City.

The Hospital Research and Ethics Committees approved the study.

### 2.1. Patients

Adult patients with severe acute respiratory syndrome coronavirus 2 (SARS-CoV-2) testing positive by polymerase chain reaction (PCR) were admitted to hospitalization and intensive care unit between 1 March 2021 and 15 March 2021. The data of 15 volunteer subjects with a negative PCR test were included as a reference group (*n* = 15).

Moderate illness was defined as mild respiratory symptoms, radiological evidence of pneumonia, and SpO2 < 90% (Group A). Severe illness was defined as SpO2 ≤ 85% requiring intubation and mechanical ventilation (Group B).

Baseline parameters such as sociodemographic variables, comorbidities, clinical symptoms, and clinical parameters were extracted from the patients’ medical records. Ventilation modes (invasive versus non-invasive), duration of illness on hospital admission (days), and medical treatment were also collected from the medical records of each patient. All included subjects were nonvaccinated.

### 2.2. Determination of Biomarkers of Pro-Inflammatory Cytokines and Endothelial Damage

Serum samples were assayed for the presence of biomarkers of inflammation (TNFα, CRP, IL-6), coagulation (Thrombin, D-dimer), oxidative stress (carbonyls and nitrites/nitrates), and Syndecan-1 as a marker of endothelial damage. All molecules were measured using commercially available kits following manufacturer instructions. CRP, D-Dimers, and IL-6 were measured by standardized central laboratory methodology. TNF-α was determined using the TNF alpha Human ELISA Kit (In Vitrogen catalog # KHC3011, Waltham, MA, USA). The measurement of thrombin was performed using the Human Thrombin ELISA Kit (Abcam, catalog # ab270210 Cambridge, U.K.). Protein carbonylation was evaluated using the Protein Carbonyl Fluorometric Assay Kit (Cayman Chemical, catalog # 701530 Ann Arbor, MI, USA). Total nitrites/nitrates concentration was determined using the Nitrate/Nitrite Fluorometric Assay Kit (Cayman Chemical, Catalog # 780051, Ann Arbor, MI, USA). Syndecan 1 (SDC1) was measured using Syndecan 1 Human ELISA Kit (Thermo Fisher Scientific, Catalog # EHSDC1, Waltham, MA, USA), Each sample was evaluated by duplicate, and the data were expressed as mean ± SD. Hemolyzed serums were discarded due to interference with the detection method.

### 2.3. Statistical Analysis

Variables are reported as mean ± standard deviation (SD) or percentages when appropriate. Data plotting and statistical analysis were performed using Prism version 7.0 for Windows software (GraphPad, CA, USA). A *p*-value <0.05 was considered statistically significant.

## 3. Results

Sixty-six patients with a clinical diagnosis and a positive PCR test for COVID-19 were included in the study. Fifteen samples from subjects without COVID-19 disease were used as controls. According to the criteria mentioned, the patients were grouped into moderate or severe illness, symptoms (Table 1), and pharmacological treatment (Table 2) of each group were recorded.

Blood chemistry values are reported in Table 3.

A significant increase in enzymes such as alkaline phosphatase, AST, ALT, and LDH was found in COVID-19 patients compared with the control group and between the moderate and severe forms, which is compatible with liver damage (Table 4).

Additionally, other alterations were found with a significant increase in PCT (suggesting bacterial concomitant infection) and ferritin, and a moderate decrease in platelets was found in the severe group (Table 4).

A D-dimer increase and a decrease in thrombin concentration in COVID-19 groups were also found (Figure 1).

In the evaluation of the inflammatory status of the patients, TNF-α, C-Reactive Protein (CRP), and Interleukin 6 (IL-6) were evaluated. Findings showed significant differences between the COVID-19 groups vs. the control group; the increase was higher in the group with severe COVID-19 (Figure 2).

Additionally, Syndecan-1 (SDC-1) was evaluated as a marker of possible microvascular endothelial damage, which was significantly increased in patients with COVID-19: 16.97 ± 5.6 ng/mL in the moderate group and 29.95 ± 8.67 ng/mL in the severe group, versus 10.29 ± 4.9 ng/mL in the control group; the increase was higher in severe cases.

We found a significant increase in nitrites/nitrates in COVID-19 patients, higher in the severe group; additionally, carbonylated proteins were increased in patients with COVID-19 vs. the control group. Both markers indicate an increase in oxidative stress (Figure 3a,b) in patients with COVID-19 disease.

Interestingly, the ratio syndecan-1/TNF-α showed that the increase in endothelial damage (measured by circulating syndecan values) was higher than the increase in TNF-α (Figure 3c). On the other hand, there was a strong correlation between the nitrite/nitrate values and the concentration of syndecan-1 (Figure 3d), suggesting a close relationship between endothelial damage and COVID-19 severity.

## 4. Discussion

COVID-19 is a lower respiratory tract infection transmitted via air droplets by symptomatic patients and asymptomatic carriers with a possible multiorgan involvement [10]. In this study, we analyzed the relationship between the severity of COVID-19 and different biomarkers for inflammation and endothelial injury with coagulopathy.

As reported [11], patients with pneumonia caused by the SARS-CoV-2 virus exhibited symptoms such as fever, dry cough, muscle and joint pain, headache, and diarrhea. COVID-19 patients may develop acute respiratory distress syndrome (ARDS), a life-threatening form of respiratory failure, around 8–9 days after symptom onset [12].

ARDS can be triggered due to direct viral effects and pro-inflammatory cytokines derived from activated host immune cells, such as IL-1, IL-6, and TNF-α, that can aggravate the patient and damage tissues such as cardiac, hepatic, and renal, resulting in multiorgan failure and death [13,14].

Increased release and activity of cytokines are derived from the severe immune response process. They are part of a phenomenon known as cytokine storm, which has been recognized as a leading cause of COVID-19 severity [15]. Significant levels of the inflammatory cytokines IL-6, IL-8, IL-10, and TNF-α have been documented in severe COVID-19 compared with non-severe disease cases, reflecting this phenomenon [16]. In our study, inflammatory cytokines TNF-a, C-Reactive Protein (CRP), and Interleukin 6 (IL-6) significantly differed between patients with moderate and severe COVID-19, indicating their potential as parameters for severity.

Similarly, some other serum biomarkers associated with an uncontrolled and dysfunctional immune response (cytokine storm) could play a predictive role in the risk, severity, and outcome of COVID-19 patients. One example is ferritin, whose serum level increases during viral infections and is associated with viral replication [17]. Our results showed that ferritin levels increased in moderate and severe groups compared with the control group; similar behavior was observed in other reports [18,19].

Pathogenesis of SARS-CoV-2 includes the death of the infected cells, activation of the innate immune response, secretion of inflammatory cytokines, and developing an oxidative stress status [19,20]. Our results showed that two indicatives of oxidative stress, i.e., nitrates/nitrites and carbonyls, are increased in patients with moderate and severe COVID-19. Protein carbonylation reflects the imbalance of production of/reduction in reactive oxygen species (ROS) due to an advanced infection process. We found no differences in the carbonylation levels between moderate and severe COVID-19. On the other hand, the nitrates/nitrites ratio showed a difference between the two stages of COVID-19 illness; this could be derived from a dysfunctional endothelium and the response of macrophages to the viral infection. Endothelial dysfunction, which seems proportional to illness severity, also relates to oxidative processes triggered by high nitric oxide levels [21]. The systemic endothelial dysfunction, in turn, reduces its ability to promote vasodilation, fibrinolysis, and anti-aggregation [22].

In searching for endothelial dysfunction biomarkers, we need to consider the glycocalyx, the first endothelial layer in contact with blood. The glycocalyx is a mixture of proteoglycans, principally syndecans and glypicans, and glycoproteins that cover the luminal surface of vascular endothelial cells and regulate the access of cells and molecules in the blood into the endothelium [23]. The electrostatic boundary between the syndecans and other extracellular matrix molecules is interrupted by inflammatory mechanisms through the activation of metalloproteinases, heparanase, and hyaluronidase by reactive oxygen species (ROS), TNF-α, and interleukin-1 beta (IL-1β) [24].

Evidence points to COVID-19 patients having higher glycocalyx-degradation products such as SDC-1 and P-selectin, which may relate to thrombosis risk in severe COVID-19 patients [25,26]. The loss of syndecan-1 from the endothelial glycocalyx leads to increased adhesion of leukocytes to endothelial cells, enhanced vascular permeability, and intravascular coagulation [27]. Our study showed that the serum SDC-1 levels were significantly higher in severe patients, suggesting a high glycocalyx degradation. SDC-1 reflects the endothelial damage directly and demonstrates the TNF-α -induced glycocalyx damage since the SDC-1/TNF-α ratio increases as the COVID-19 severity increases.

The values of SDC-1 and nitrites/nitrates show a high correlation level, also suggesting a close relationship between endothelial dysfunction and COVID-19 severity.

On the other hand, PCT is usually produced in response to bacterial infections, but increasing evidence shows that elevated levels are associated with a higher risk of severe SARS-CoV-2 disease [28]. A possible explanation for the amplified PCT involves the cytokine storm due to viral sepsis in severe and critical cases of COVID-19 [29].

Interestingly, D-dimer, a fibrin degradation product, is widely used as a biomarker for thrombotic disorders, and elevated concentrations are found in patients with severe COVID-19 [30]. They may be related to the progression of the disease and poor outcomes [31]. Our results showed a significantly higher D-dimer concentration in the severe group than in the moderate group.

Although we did not observe a correlation between cardiovascular comorbidities and a poorer prognosis, previous studies have shown that they are risk factors that contribute to a greater mortality among COVID-19 patients [32]. Recent evidence suggests that circulating miRNAs in COVID-19 patients are associated with multiorgan failure, thrombosis, and death [33]. The potential long-term consequences on the cardiovascular system of recovered patients are unknown.

## 5. Conclusions

In summary, the present findings provide further support for using SDC-1 alone or combined with other markers such as IL-6, TNF-α, CRP, PCT, and D-dimer as possible indicators for predicting severity and could be a valuable approach for monitoring the disease activity of COVID-19. Future studies should consider drugs that act as pro-inflammatory mediators or protect or restore the endothelial glycocalyx, particularly in long COVID.

## Figures and Tables

**Figure 1 jcm-11-07436-f001:**
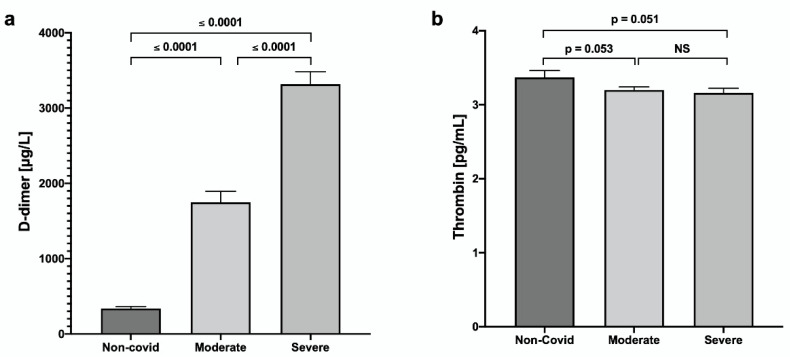
Evaluation of serum concentrations of (**a**) D-dimer and (**b**) thrombin in patients with COVID-19. Values are expressed as mean ± SE.

**Figure 2 jcm-11-07436-f002:**
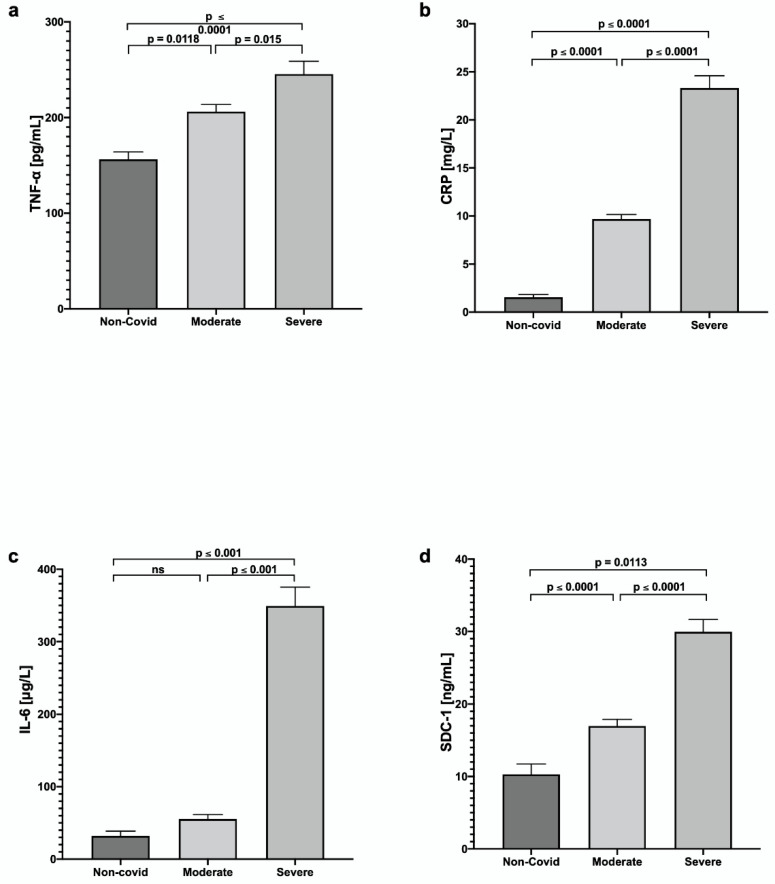
Analysis of the systemic inflammatory state of patients with moderate and severe COVID-19 vs. control: (**a**) Tumor Necrosis Factor-alpha (TNF-α), (**b**) C-Reactive Protein (CRP), (**c**) Interleukin 6 (IL-6), and (**d**) Syndecan-1 (SDC-1). Values are expressed as mean ± SE.

**Figure 3 jcm-11-07436-f003:**
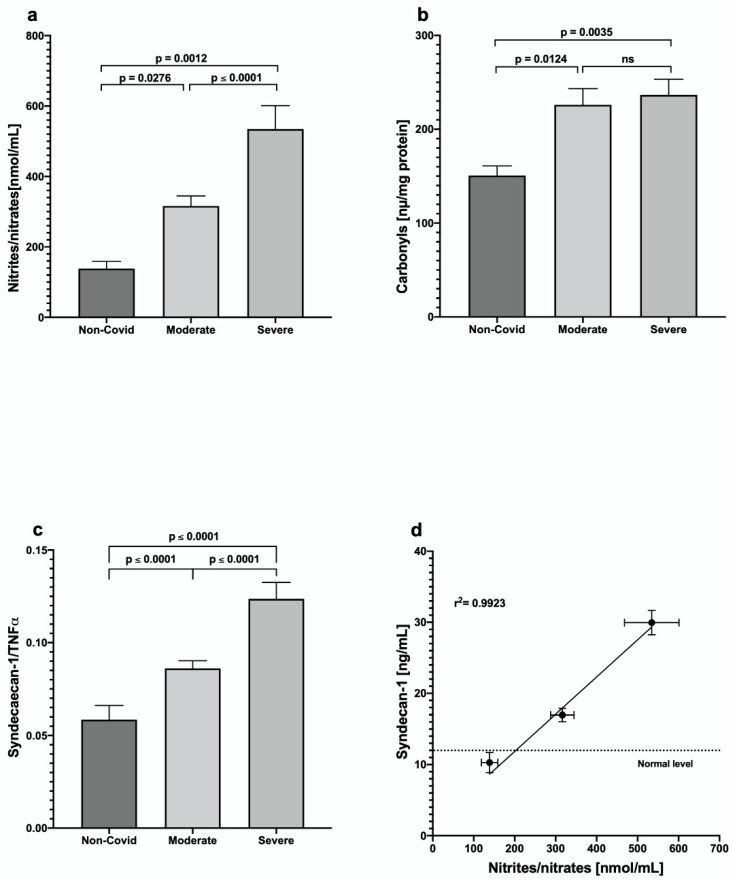
Evaluation of the oxidative status of patients with COVID-19 disease: (**a**) Nitrites/nitrates, (**b**) Carbonyls, (**c**) Syndecan-1/TNF-α and (**d**) Syndecan-1. Values are expressed as means ± SE.

**Table 1 jcm-11-07436-t001:** Symptoms associated with COVID-19 in patients with moderate and severe disease.

	Non-COVID-19 *n* = 15	Moderate *n* = 37	Severe *n* = 29
Age (years)	53.13 ± 90	60.49 ± 14.62	65.42 ± 12.07
Symptoms, *n* (%)			
Fever	0	19 (51.35)	16 (55.17)
Cephalea	1	14 (37.83)	7 (24.13)
Congestion	0	2 (5.40)	1 (3.44
Rhinorrhea	0	8 (21.62)	3 (10.34)
Cough	0	23 (62.16)	21 (72.41)
Odynophagia	0	6 (16.21)	6 (20.68)
Arthralgias	0	25 (67.56)	13 (44.82)
Myalgia	0	23 (62.16)	13 (44.82)
Dyspnea	0	30 (8.10)	26 (89.65)
Diarrhea	0	5 (13.51)	5 (17.24)

**Table 2 jcm-11-07436-t002:** Pharmacological treatment of patients with COVID-19 disease.

Characteristics	Moderate *n* = 37	Severe *n* = 29
Days of treatments *	21.76 ± 13.25	23.42 ± 11.26
Prone positioning, *n* (%)	11 (29.72)	23 (79.31)
LOMV, median [IQR] **	0.00	15.23 ± 11.15
ICU LOS, median [IQR] **	0.30 ± 1.81	5.81 ± 9.2
Hospital LOS median [IQR] **	21.73 ± 13.25	22.81± 11.20
Comorbidities, *n* (%)		
Diabetes mellitus	16 (43.24)	8 (27.58)
Hypertension	13 (35.13)	8 (27.58)
Obesity	4 (10.81)	3 (10.34)
Chronic kidney disease	1 (2.70)	0
COVID-19 specific treatments *n* (%)		
Azithromycin	12 (32.43)	9 (31.03)
Ceftriaxone	13 (35.13)	15 (51.72)
Levofloxacin	13 (35.13)	12 (41.37)
Clarithromycin	1 (2.70)	0
Budesonide	5 (13.51)	6 (20.68)
Dexamethasone	26 (70.27)	24 (82.75)
Enoxaparin	30 (81.08)	22 (75.86)
Ivermectin	11 (29.72)	7 (24.13)
Methylprednisolone	5 (13.51)	15 (51.72)
Hydrocortisone	2 (5.40)	2 (6.89)
Prednisone	2 (5.40)	0

LOMV = length of mechanical ventilation; ICU = intensive Care Unit; LOS = length of stay. * mean ± SD; ** median ± SD.

**Table 3 jcm-11-07436-t003:** Parameters of serum evaluations. Values represent mean ± standard deviation.

Parameter	Non-COVID-19	Moderate	Severe
Glucose (mg/dL)	149.75 ± 43.00	160.24 ± 70.94	179.95 ± 49.51
Urea (mg/dL)	65.80 ± 6.42	42.07 ± 16.28	65.56 ± 20.08
Creatinine (mg/dL)	2.37 ± 0.47	0.85 ± 0.28	1.48 ± 0.54
Uric acid (mg/dL)	6.60 ± 1.78	5.33 ± 1.66	7.01 ± 1.13
Cholesterol (mg/dL)	155.13 ± 34.71	152.3 ± 45.70	142.58 ± 33.51
Triglycerides (mg/dL)	139.25 ± 34.20	166.21 ± 60.14	161.17 ± 52.34
HDL-c (mg/dL)	33.73 ± 4.22	36.44 ± 6.34	38.21 ± 6.29
LDL-c (mg/dL)	96.05 ± 25.15	103.33 ± 33.41	96.19 ± 32.79
BUN (mg/dL)	22.44 ± 3.85	21.55 ± 8.60	30.39 ± 12.34

**Table 4 jcm-11-07436-t004:** Biochemical values of the groups studied. Values represent mean ± SD.

Parameter	Non-COVID-19	Moderate	*p* ^#^	Severe	*p*
DBIL, mg/dL	0.28 ± 0.07	0.34 ± 0.12	NS	0.38 ± 0.18	NS
IBIL, mg/dL	0.40 ± 0.05	0.32 ± 0.11	NS	0.27 ± 0.12	0.0100 ^#^
Total bilirubin, mg/dL	0.69 ± 0.07	0.66 ± 0.26	NS	0.65 ± 0.28	NS
ALT, U/L	27.65 ± 5.54	69.03 ± 22.06	0.0001	42.27 ± 23.95	0.0001 ^##^
AST, U/L	40.53 ± 2.82	66.20 ± 10.04	0.0008	44.64 ± 24.85	0.0001 ^##^
Alkaline phosphatase, U/L	154.41 ± 25.46	109.04 ± 37.68	0.0060	131.14 ± 37.87	0.0100 ^#^
GGT, U/L	146.05 ± 34.83	130.24 ± 41.82	NS	175.27 ± 37.09	0.0001 ^##^
LDH, U/L	294.00 ± 31.11	377.55 ± 81.68	0.0001	601.10 ± 182.62	0.0001 ^##^
Total Protein, mg/dL	5.70 ± 0.95	6.37 ± 0.85	NS	5.91 ± 0.69	NS
Albumin, g/dL	3.18 ± 0.71	3.59 ± 0.53	NS	3.19 ± 0.58	0.0100 ^##^
Globulin, g/dL	2.60 ± 0.59	2.78 ± 0.59	NS	2.70 ± 0.33	NS
PCT, ng/mL	0.36 ± 0.16	0.27 ± 0.10	NS	2.75 ± 0.44	0.0001 ^#^ 0.0001 ^##^
Platelets, per mL	244.00 ± 46.29	250.20 ± 94.87	NS	214.71 ± 43.46	NS ^#^ NS ^##^
Ferritin, mg/L	365.88 ± 56.05	924.69 ± 270.81	0.0001	858.07 ± 257.37	0.0001 ^#^ NS ^##^

^#^ *p*-value, versus non-COVID-19 group, ^##^ *p*-value, severe vs. moderate.

## Data Availability

Not applicable.

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
