# Peer review of "Correlation of Biomarkers of Endothelial Injury and Inflammation to Outcome in Hospitalized COVID-19 Patients"

_jcm, 2022, doi:10.3390/jcm11247436_

Round 1
Reviewer 1 Report
This study aimed to elucidate whether the biomarkers of inflammation and endothe-66 lial damage are related to clinical severity and outcome of patients with COVID-19 and to identify patients who may develop long-term consequences and require close clinical monitoring in rehabilitation.
- The topic is important and interesting in the context of the current literature.
- Methods are well described and appropriate.
- Results are well presented.
- Discussion needs to be expanded a little bit, citing more pertinent references on this topic such as : Ielapi N, et al. Cardiovascular disease as a biomarker for an increased risk of COVID-19 infection and related poor prognosis. Biomark Med. 2020 Jun;14(9):713-716.
Author Response
Munguia et al- jcm-2068125-R1
Reviewer #1
This study aimed to elucidate whether the biomarkers of inflammation and endothelial damage are related to clinical severity and outcome of patients with COVID-19 and to identify patients who may develop long-term consequences and require close clinical monitoring in rehabilitation.
- The topic is important and interesting in the context of the current literature.
- Methods are well described and appropriate.
- Results are well presented.
- Discussion needs to be expanded a little bit, citing more pertinent references on this topic such as : Ielapi N, et al. Cardiovascular disease as a biomarker for an increased risk of COVID-19 infection and related poor prognosis. Biomark Med. 2020 Jun;14(9):713-716.
Thank you for your comments, we include the article you ask for in the discussion
Reviewer 2 Report
The manuscript by Levy Munguia et al. entitled “Correlation of biomarkers of endothelial injury and inflammation to outcome in hospitalized COVID-19 patients” aimed to elucidate whether the biomarkers of inflammation and endothelial damage are related to clinical severity and outcome of patients with COVID-19 and to identify patients who may develop long-term consequences and require close clinical monitoring in rehabilitation.
I read with great interest this paper. However, some major issues need to be addressed to improve the significance of the manuscript:
-Firstly, the abstract is not well organized and it does not summarize the general significance of the manuscript.
- Cardiovascular (CV) involvement is a crucial complication in COVID-19, and no strategies are available to prevent or specifically address CV events in COVID patients. The identification of molecular partners contributing to CV manifestations in COVID-19 patients is crucial for providing early biomarkers, prognostic predictors and new therapeutic targets. Specifically, miRNAs have been proposed as valuable biomarkers and predictors of both cardiac and vascular damage occurring in SARS-CoV-2 infection. Consequently, this article could be discussed: Izzo C, Visco V, Gambardella J, et al. Cardiovascular implications of miRNAs in COVID-19 [published online ahead of print, 2022 Jul 2]. J Pharmacol Exp Ther. 2022;JPET-MR-2022-001210. doi:10.1124/jpet.122.001210.
-Another essential point is COVID-19 vaccination: how many of the enrolled patients had already had the COVID-19 vaccine? As expected, vaccinated patients should have a better prognosis.
- Moreover, the different comorbidities of patients should be reported.
-Finally, extensive editing of English language and style is required.
Author Response
Munguia et al- jcm-2068125-R1
Reviewer #2
The manuscript by Levy Munguia et al. entitled “Correlation of biomarkers of endothelial injury and inflammation to outcome in hospitalized COVID-19 patients” aimed to elucidate whether the biomarkers of inflammation and endothelial damage are related to clinical severity and outcome of patients with COVID-19 and to identify patients who may develop long-term consequences and require close clinical monitoring in rehabilitation.
I read with great interest this paper. However, some major issues need to be addressed to improve the significance of the manuscript:
-Firstly, the abstract is not well organized and it does not summarize the general significance of the manuscript.
Thank you for your comments we rewrite the abstract as you suggested now reads as follows:
COVID-19 can trigger an intense systemic inflammation and prothrombotic state, leading to a rapid and disproportionate deterioration of lung function. An effective screening tool is essential to identify the patients at risk for severe disease. This observational study was conducted on hospitalized patients with moderate and severe COVID-19 pneumonia in a General Hospital in Mexico City between March 1 and 15, 2021. Serum samples were analyzed to explore the role of biomarkers of inflammation, coagulation, oxidative stress, and endothelial damage with the severity of the disease. Our results demonstrated that Syndecan-1 and nitrites/nitrates showed a high correlation in severely ill patients. In conclusion, COVID-19 patients with elevated levels of SDC-1 were associated with severe disease. This molecule can potentially be used as a marker for the progression or severity of COVID-19. Preservation of glycocalyx integrity may be a potential treatment for COVID-19.
- Cardiovascular (CV) involvement is a crucial complication in COVID-19, and no strategies are available to prevent or specifically address CV events in COVID patients. The identification of molecular partners contributing to CV manifestations in COVID-19 patients is crucial for providing early biomarkers, prognostic predictors and new therapeutic targets. Specifically, miRNAs have been proposed as valuable biomarkers and predictors of both cardiac and vascular damage occurring in SARS-CoV-2 infection. Consequently, this article could be discussed: Izzo C, Visco V, Gambardella J, et al. Cardiovascular implications of miRNAs in COVID-19 [published online ahead of print, 2022 Jul 2]. J Pharmacol Exp Ther. 2022;JPET-MR-2022-001210. doi:10.1124/jpet.122.001210.
Thank you for your comments, we include the reference you suggested in the discussion section
-Another essential point is COVID-19 vaccination: how many of the enrolled patients had already had the COVID-19 vaccine? As expected, vaccinated patients should have a better prognosis.
Thank you for your comments, we agree with you. All included subjects were nonvaccinated at the study time. We include this statement in the text
- Moreover, the different comorbidities of patients should be reported.
Thank you for your comments, we now include comorbidities in table 2
-Finally, extensive editing of English language and style is required.
Thank you for your comments, an extensive English language edition was performed
Round 2
Reviewer 2 Report
The paper could be accepted in present form